# PROVABLY CALIBRATED REGRESSION UNDER DISTRIBUTION DRIFT

## ABSTRACT

Accurate uncertainty quantification is a key building block of trustworthy machine learning systems. Uncertainty is typically represented by probability distributions over the possible outcomes, and these probabilities should be calibrated, *e.g.* the 90% credible interval should contain the true outcome 90% of the times. In the online prediction setup, existing conformal methods can provably achieve calibration assuming no distribution shift; however, the assumption is difficult to verify, and unlikely to hold in many applications such as time series prediction. Inspired by control theory, we propose a prediction algorithm that guarantees calibration even under distribution shift, and achieves strong performance on metrics such as sharpness and proper scores. We compare our method with baselines on 19 time-series and regression datasets, and our method achieves approximately 2x reduction in calibration error, comparable sharpness, and improved downstream decision utility.

## 1 INTRODUCTION

Accurate uncertainty quantification is crucial for machine learning predictions used in high-stakes decision making. Typically, uncertainty is represented by probability distributions over the possible outcomes, and these probabilities should be calibrated. In the regression setup, for example, the true label should be below the predicted 95% quantile for 95% of the samples (Gneiting et al., 2007). Calibration can convey confidence to decision makers because extreme values (e.g. true label above 95% quantile) are guaranteed to be rare. In addition to calibration, these probabilities should be sharp (i.e. concentrated and have low variance). Sharp probabilities are useful to decision makers because they are informative.

If data is i.i.d., then recalibration (Kuleshov et al., 2018) and conformal prediction (Vovk et al., 2020) algorithms can achieve low calibration error and good sharpness in the regression setup. However, the i.i.d. assumption is unlikely to hold in most time-series prediction tasks. Under distribution drift, it is possible to adapt regret minimization algorithms (Cesa-Bianchi & Lugosi, 2006; Kuleshov & Ermon, 2017) to achieve calibration. However, regret minimization calibration algorithms are designed for the asymptotic regime, and empirically we show that they are effective only with large sample size, and have very poor calibration and sharpness with short time series (e.g. 50 samples), limiting their practical utility.

Our goal is to design an algorithm to achieve good calibration and sharpness in the online regression setup even for short time series. We start with an existing prediction algorithm that has good calibration and sharpness for i.i.d. data (such as conformal prediction), and track the empirical frequency that the labels are below e.g. the 75% quantile. If the empirical frequency is significantly below 75%, we move future predictions up so more labels are below the prediction; vice versa. We design a very efficient adjustment method that can guarantee near perfect calibration with only tens of samples. We call this the "basic" prediction algorithm shown in Figure 1 (upper right).

Our main technical contribution is an improved algorithm that addresses two short-comings of the "basic" algorithm without breaking guaranteed calibration. First, the predictions should be feasible, e.g. the 75% quantile should never be smaller than the 25% quantile. The basic algorithm might violate this requirement because we adjust each quantile separately with no constraint on their relation. Second, the predictions should be stable if the distribution drift stops; empirically instability harms sharpness and proper scores. Our improved algorithm is based on an analogy between the

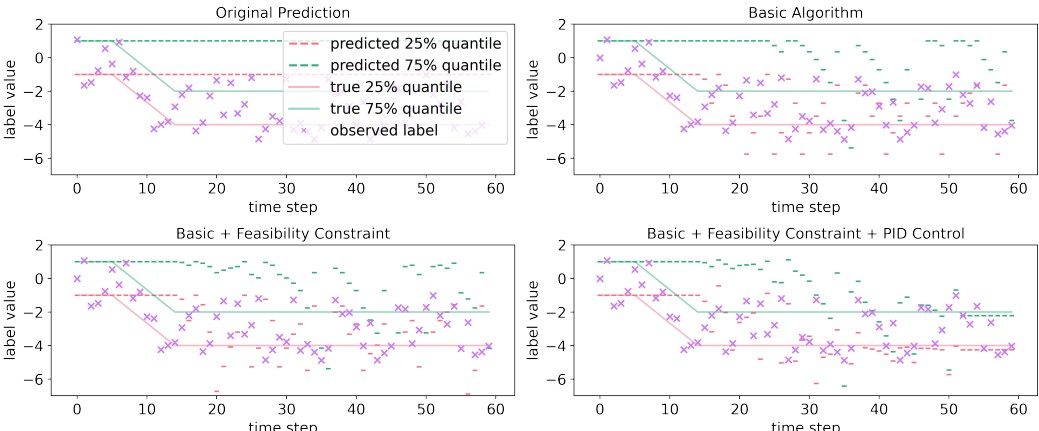

Figure 1: An illustrative example of our algorithm. The $x$-axis is the time step (for the online prediction setup), and the $y$-axis is the prediction. We look at the 75% and the 25% quantile of the prediction. The purple markers are the true labels. The true label is initially drawn from $\mathrm{Uniform}[-2, 2]$ but drifts to $\mathrm{Uniform}[-5, -1]$. **Upper Left**: The original predictions that is not calibrated because distribution drifted at time step 5. **Upper Right**: The "basic" algorithm: we move the quantiles up or down to achieve calibration. However, the prediction is sometimes infeasible (e.g. the 25% quantile is above the 75% quantile at time step 43) and oscillates even when distribution drift stops. **Lower Left**: We add feasibility constraints. The quantiles are calibrated and feasible. **Lower Right**: We use PID control to stabilize the predictions. The quantiles are calibrated, feasible and do not oscillate. The predictions adapts very quickly in response to a new distribution within tens of samples.

prediction task and a mechanical system. We encode the feasibility constraints as the feasible states of a mechanical system, and use proportional-integral-derivative (PID) control (Minorsky, 1922) to stabilize the system. Our final algorithm makes predictions that satisfy all three desirable properties: provable calibration, feasibility, and stability (Figure 1 bottom right).

We test our algorithm on two real world time series datasets (stock earnings and COVID cases) and a large benchmark of 17 common regression datasets. We reduce calibration error by approximately 2x compared too all baselines, while achieving comparable or better sharpness. We also simulate COVID response decisions based on predictions on the COVID dataset, and observe improved decision loss compared to baselines during periods of distribution drift (e.g. when case numbers surge).

## 2 PROBLEM SETUP AND BACKGROUND

We consider regression problems. Let $X \in \mathcal{X}$ denote the input feature and $Y$ denote the label. We assume that the label is bounded by $B$, i.e. $Y$ cannot take any value outside $[-B, B]$ for some pre-specified $B > 0$.

We consider quantile predictions with a set of $K$ equally spaced quantiles $0 < \alpha_1 < \cdots < \alpha_K < 1$. For example, when $K = 9$ we choose $\alpha_1 = 10\%, \alpha_2 = 20\%, \cdots, \alpha_K = 90\%$. A quantile prediction is a vector of $K$ numbers (denoted by $Z$) where $Z_k$ should ideally predict the $\alpha_k$-th quantile of the label $Y|X$ conditioned on the input feature $X$. We say a quantile prediction is feasible, if $Y_k < Y_{k+1}, \forall k$, i.e. a lower quantile is always smaller than a higher quantile. In the limit of infinitely many quantiles $K \to \infty$, a quantile prediction is equivalent to a probability distribution, however, for technical reasons that will become evident later, we only consider finitely many quantiles in this paper.

Our setup is an online prediction setup, where the forecaster sequentially makes the predictions. Consider a COVID prediction example: at time $t$, the forecaster observes some features $X^t$ (such as current vaccination rate) and makes a prediction $Z^t$ about next week's case number. After the forecaster makes a prediction, the true case number $Y^t$ is revealed; then we move on to time step

$t + 1$ and repeat this process. Following standard terminology we will use "nature" to refer to the process that generates the features $X^t$ and the true label $Y^t$. Our setup is formalized by the following interaction between forecaster and nature: for $t = 1, 2, \cdots, T$

1. Nature reveals feature $X^t$
2. Forecaster makes a quantile prediction $Z^t \in \mathbb{R}^K$ where $Z_k^t$ is the $\alpha_k$-th quantile.
3. Nature reveals label $Y^t \in [-B, B]$

We call a finite sequence of interactions a transcript, i.e. a sequence $X^1, Z^1, Y^1, \cdots, X^T, Z^T, Y^T$. A forecasting algorithm $\psi$ is a function that maps a transcript and a new feature to a prediction:

$$\psi : X^1, Z^1, Y^1, \cdots, X^{t-1}, Z^{t-1}, Y^{t-1}, X^t \mapsto Z^t. \tag{1}$$

We will consider two types of assumptions on nature. We say that nature is **i.i.d.** if $\forall t$, $X^t, Y^t$ are drawn from some i.i.d. random variable (but we make no assumption on the probability law on this random variable). We say that nature is **causal** if nature chooses $X_t, Y_t$ without depending on the future. Specifically, $X^t, Y^t$ can only depend on the variables that precede it in the transcript (i.e. $X^t$ can depend on $X^1, \cdots, Y^{t-1}$ and $Y^t$ can only depend on $X^1, \cdots, Y^{t-1}, X^t, Z^t$). Causal is an extremely weak assumption. For example, nature can even adversarially choose the label $Y^t$ to increase prediction error after observing the forecaster's prediction $Z^t$.

### 2.1 Calibration and Achieving Calibration with Conformal Methods

A natural property that we can request is (probabilistic) calibration (Gneiting et al., 2007; Kuleshov et al., 2018). Intuitively, $\forall k$, the label $Y^t$ should be below the predicted $\alpha_k$-th quantile for an $\alpha_k$ proportion of the samples. Formally, given a transcript of any length, consider the empirical frequency that the label $Y^t$ is below the predicted quantile $Z_k^t$ up to time $T$ (for any $T$ that's less than the length of transcript)

$$F_k^T = \sum_{1 \leq t \leq T} \mathbb{I}(Y^t \leq Z_k^t). \tag{2}$$

Ideally, the label $Y^t$ should be below the $\alpha_k$-th quantile exactly $\alpha_k T$ many times, i.e., $\forall k, F_k^T / T = \alpha_k$. Achieving this perfectly is difficult, so we define an approximation. For any function $b : \mathbb{N} \to \mathbb{R}^+$ we say that a transcript is $b$-calibrated if $\forall k = 1, \cdots, K, \forall T, |F_k^T / T - \alpha_k| \leq b(T)$. Correspondingly, a forecasting strategy is calibrated if the resulting transcript is calibrated:,

**Definition 1.** *For any function $b : \mathbb{N} \to \mathbb{R}^+$, a forecasting algorithm $\psi$ is $b$-calibrated under i.i.d. (or causal) assumptions if for any nature's strategy that is i.i.d. (or causal), the resulting transcript is $b$-calibrated almost surely.*

**Conformal Calibration** Conformal calibration (Vovk et al., 2020) is a forecasting algorithm based on conformal prediction ideas (Vovk et al., 2005; Shafer & Vovk, 2008). It is a wrapper algorithm that transforms a initial predictor (such as an off-the-shelf predictor) into new predictions that are provably calibrated under i.i.d. assumptions. Conformal calibration is based on the following intuition: for example, if 75% of past labels are below the initial prediction's mean, then we can use the initial prediction's mean as the 75%-quantile prediction in the future. More generally, we identify a statistic of the initial prediction, such that $\alpha_k$-proportion of the past labels are below that statistic. We use this statistic as the $\alpha_k$-th quantile of the future prediction.

If the data is i.i.d., conformal calibration is a very useful algorithm. This is because it has extremely strong calibration properties (Proven in Proposition 1 of (Vovk et al., 2020)), where the random variables $\mathbb{I}(Y^t \leq Z_k^t), t = 1, 2, \cdots$ is a sequence of i.i.d. Bernoulli random variables with mean $\alpha_k$. This is impressive because even if we have access to the oracle forecaster (i.e. $Y_k^t$ is equal to the true conditional $\alpha_k$-th quantile of $Y^t | X^t$), we cannot achieve better calibration — $\mathbb{I}(Y^t \leq Z_k^t)$ is also a sequence of i.i.d. Bernoulli random variables with mean $\alpha_k$. Empirically, conformal prediction also has very good sharpness if the initial prediction function is reasonable (Burnaev & Vovk, 2014).

A minor property (but needed for proofs) is that the predictions should bounded in $[-B, B]$. Conformal calibration satisfies this property when $t > K$, i.e. there are more samples than predicted quantiles. To bypass this limitation, we can initialize the conformal calibration algorithm with at least $K$ offline samples. Alternatively, if we know $B$ in advance, we can clip the prediction by $B$.

# 3 CALIBRATION UNDER DISTRIBUTION DRIFT

This section introduces our algorithm that guarantees calibration even when nature is not i.i.d. We start from a basic algorithm in Section 3.1, and discuss its shortcomings. Then we move on to more advanced algorithms in Section 3.2 and 3.3 to address the shortcomings.

## 3.1 A BASIC CALIBRATED PREDICTION ALGORITHM

Conformal calibration achieves good calibration and sharpness when nature is i.i.d. When we are unsure if nature is i.i.d., our basic idea is to use conformal calibration until it has failed to achieve calibration. Specifically, when nature is i.i.d., $\mathbb{I}(Y^t \leq Z_k^t)$ is a sequence of Bernoulli random variables, so $F_k^T$ is the sum of these Bernoulli random variables (i.e. a binomial). We use $b_\delta^*(T)$ to denote the confidence interval of a binomial distribution, i.e. $\Pr[F_k^T/T \in \alpha_k \pm b_\delta^*(T)] = 1 - \delta$. We fix some small $\delta$ (such as 0.05) and if under the conformal calibration algorithm $F_k^T/T \notin \alpha_k \pm b_\delta^*(T)$, we know (with 0.95 confidence) that nature is not i.i.d, so we will make adjustments to salvage calibration.

From a high level, $F_k^t > \alpha_k t + b_\delta^*(t)t$ implies that the true label is below the $\alpha_k$-th quantile too often, so our prediction $Z_k^t$ has been too large and we should reduce it; vice versa; There are some design freedom in choosing how much to reduce or increase the prediction. We choose an adjustment that grows exponentially with larger calibration error. The reason for this choice is to make very large adjustments when the calibration error is large, so we can tightly bound the calibration error. Formally, let $\tilde{Z}_k^t$ denote the initial prediction generated by the conformal calibration prediction algorithm, the prediction of our algorithm $Y_k^t$ is given by

$$Z_k^{t+1} = \tilde{Z}_k^{t+1} + E_k^{t+1}, \quad \text{where } E_k^{t+1} = \begin{cases} 1 - e^{\beta(F_k^t - \alpha_k t - b_\delta^*(t)t)} & F_k^t > \alpha_k t + b_\delta^*(t)t \\ e^{\beta(\alpha_k t - b_\delta^*(t)t - F_k^t)} - 1 & F_k^t < \alpha_k t - b_\delta^*(t)t \\ 0 & \text{otherwise} \end{cases} \quad (3)$$

Intuitively, if $F_k^t$ is within the correct range we make no adjustments (i.e. $Z_k^{t+1} = \tilde{Z}_k^{t+1}$); when $F_k^t$ falls outside the correct range, we make aggressive adjustments that increase exponentially with how much it falls outside the correct range. $\beta > 0$ is a hyper-parameter that controls how large the adjustments are (we will show in Section 3.4 that these hyper-parameters are easy to choose). Note that $E_k^{t+1}$ is for the $(t + 1)$-th time step (instead of $t$). This is because any adjustments can only depend on past information, so we will only have access to $F_k^t$ at the $(t + 1)$-th time step. The following theorem shows that such adjustments indeed guarantee calibration.

**Theorem 1.** *Let $\tilde{Z}_k^t$ be generated by any forecasting algorithm and bounded in $[-B, B]$, for any $\delta > 0$, the prediction algorithm defined by Eq.(3) is b-calibrated where $b(T) = b_\delta^*(T) + \frac{\log(2B+1) + \beta}{\beta T}$.*

Note that the guarantee of Theorem 1 does not require that conformal calibration generate the initial prediction $\tilde{Z}_k^t$. The use of conformal calibration is motivated by its strong empirical performance (Vovk et al., 2020) when the data is i.i.d. or close to i.i.d. (which is further supported by our experiments).

The calibration guarantee is strong because under conformal calibration with i.i.d. data (or the oracle forecaster) we can expect to see $b_\delta^*(T)$ calibration error (with $1 - \delta$ probability); without i.i.d. assumptions, our algorithm can ensure that the calibration error only increases by $O(\log B/T)$. Notably, the calibration error scales logarithmically with the assumed upper bound $\log B$, so choosing a loose bound $B$ does not significantly degrade the guarantee.

The basic idea has two shortcomings illustrated in Figure 1 that we will address in the next sections. First, in a feasible quantile prediction, the 75% quantile should never be smaller than the 25% quantile, but our adjustments might violate this requirement because we adjust each quantile separately with no constraint. Second, the basic idea might lead to oscillation and instability. Intuitively, when the empirical frequency is incorrect, we make adjustments to correct it, but once the empirical frequency improves, we reduce the adjustments, which will cause the empirical frequency to become incorrect again, leading to oscillation. Empirically instability can hurt performance metrics such as sharpness.

## 3.2 IMPROVING THE BASIC ALGORITHM TO ENSURE FEASIBILITY

In this section, we address the feasibility problem. We first define some convenient notation shorthand. For our prediction $Z^t$, we denote the distance between (neighboring) quantiles as $\Delta Z_k^t = Z_{k+1}^t - Z_k^t$ for $k = 0, \cdots, K$. We can make a similar definition for the initial prediction $\Delta \tilde{Z}_k^t = \tilde{Z}_{k+1}^t - \tilde{Z}_k^t$. Under these new definitions, requiring feasibility is equivalent to requiring positive distance between neighboring quantiles, i.e. $\Delta Z_k^t > 0, \forall k$.

The main challenge is that the basic adjustment algorithm Eq.(3) might not satisfy this new requirement, so we need to find a compromise. We draw inspiration from a mechanical system illustrated in Figure 2 where we imagine that our prediction quantiles $Z_1^t, \cdots, Z_K^t$ are a sequence of points in space. We imagine that each requirement (satisfying Eq.(3) or feasibility constraint) can be thought of as a "force" that "push" the quantiles to satisfy the requirement. These forces will contend with each other (because the requirements are conflicting), so the equilibrium of these forces is the natural compromise between the conflicting requirements.

Specifically, to enforce the requirement that the interval size $\Delta Z_k^t$ cannot be close to 0 or negative, we design a force that pushes the interval to be larger whenever it is small. Intuitively, we can think of this as inserting a "spring" (illustrated in Figure 2) into the interval. The spring will resist being compressed, and exert an opposing force whenever the interval becomes small. The smaller the interval, the more compressed the spring will be, so the larger the opposing force; in the limit where the interval size tends to 0, the opposing force will tend to $\infty$. This prevents the interval size from going to zero (or negative) and ensures feasibility.

We add another desiderata: no force is generated only when the interval size $\Delta Z_k^t$ equals the initial prediction's size $\Delta \tilde{Z}_k^t$. Intuitively, we want to use the initial conformal prediction as much as possible (which we argued in Section 3.1 is desirable). This can be achieved by imaging that the spring has initial length $\Delta \tilde{Z}_k^t$ and resists both compressing and stretching.

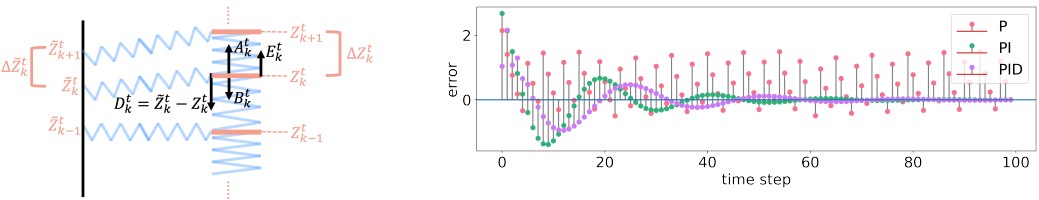

Figure 2: **Left**: An illustration for Section 3.2 where we draw analogy to a mechanical system. We attach a "spring" between neighboring quantiles that resists being compressed or stretched. The final prediction $Z_k^t$ is the equilibrium state of this mechanical system. This is the natural way to compute the compromise of several conflicting requirements. **Right**: Illustration of the effect of PID in Section 3.3. Here we plot the error $F_k^t - \alpha_k t$ when the initial prediction systematically underpredicts. P corresponds to the basic algorithm, and the error oscillates around 0. PI corresponds to adding an integral term, and the error converges to 0. PID adds an additional derivative term, which reduces overshoot and improves convergence.

To ensure that no prediction exceeds the bounds $[-B, B]$, we also create two additional quantiles $\alpha_0 = 0$ and $\alpha_{K+1} = 1$, and define $Z_0 = -B$ and $Z_{K+1} = +B$, so that assuming feasibility we will always have $Z_1 > Z_0 = -B$ and $Z_K < Z_{K+1} = B$. We formalize the above intuition as the following list of forces on each quantile:

1. Forces from miscalibration $E_k^t$ defined by Eq.(3) and deviation from initial prediction $D_k^t := Z_k^t - \tilde{Z}_k^t$. If we only have these two forces, we recover the basic algorithm. This is because if we solve the equilibrium state of the system $D_k^t + E_t^k = 0$, we get $Z_k^t = \tilde{Z}_k^t + E_k^t$ which is exactly Eq.(3).

2. Force from the spring in the interval above $A_k^t := \eta(\Delta Z_k^t / \Delta \tilde{Z}_k^t - \Delta \tilde{Z}_k^t / \Delta Z_k^t)$ where $\eta$ is a hyper-parameter (we will show in Section 3.4 that setting these parameters is easy). Intuitively, if the interval above is larger than its initial size (i.e. $\Delta Z_k^t > \Delta \tilde{Z}_k^t$) then the

spring "stretches" which generates an upward force $A_k^t > 0$; conversely if the interval above is smaller than its initial size (i.e. $\Delta Z_k^t < \Delta \tilde{Z}_k^t$) then the spring above "compresses" which generates a downward force $A_k < 0$. If the interval has the same size, then no force is generated because $A_k^t = 0$.

3. Force from the spring in the interval below $B_k^t := \eta(\Delta \tilde{Z}_{k-1}^t / \Delta Z_{k-1}^t - \Delta Z_{k-1}^t / \Delta \tilde{Z}_{k-1}^t)$.

We define the final prediction $Z_k^t$ as the solution to the following equilibrium equations (i.e. all forces sum to 0 for all the quantiles)

$$E_k^t + D_k^t + A_k^t + B_k^t = 0, \forall k = 1, \cdots, K. \tag{4}$$

The resulting forecasting algorithm is provided as Algorithm 1.

---

**Algorithm 1** Basic + Feasibility Forecasting Algorithm

---

1: For each $k = 1, \cdots, K$, initialize $F_k^0 = 0$
2: **for** $t = 1, 2, \cdots$ **do**
3:     Observe $X^t$ and use initial prediction algorithm (e.g. conformal calibration) to predict $\tilde{Z}^t$.
4:     For each $k = 1, \cdots, K$, compute $E_k^t$ according to Eq.(3).
5:     Use a physics simulation to find $Z_k^t$ that solves Eq.(4) given $\tilde{Z}^t$ and $E_k^t$, output $Z_k^t$.
6:     Observe the true label $Y^t$. For each $k = 1, \cdots, K$, set $F_k^t = F_k^{t-1} + \mathbb{I}(Y^t \le Z_k^t)$.
7: **end for**

---

**Calibration Properties** We will show that Algorithm 1 achieves calibration except when the "likelihood" is exponentially high (which is also very desirable). Specifically, we define a notion similar to likelihood

$$L(Z^t, Y^t) = (\alpha_{k+1} - \alpha_k)/\Delta Z_k^t, \text{ for the } k \text{ that satisfies } Y^t \in [Z_k^t, Z_{k+1}^t]. \tag{5}$$

Intuitively, $L(Z^t, Y^t)$ is large if the true label $Y^t$ belongs to a small interval (i.e. $\Delta Z_k^t$ is small). In the limit of infinitely many quantiles $K \to \infty$, the quantile prediction becomes equivalent to a probability distribution, and Eq.(5) becomes the likelihood of the probability distribution.

**Theorem 2** (Simplified). *Under the same condition as Theorem 1, for any $c > 2/\beta$, Algorithm 1 is b-calibrated where $b = O(K/\sqrt{T} + c/T)$ on any transcript such that $\forall t, L(Z^t, Y^t) \le e^{\Theta(c)}$.*

Theorem 2 is a simplified version of the full theorem (we replace all the constants with asymptotics). For a non-asymptotic version and proof see Appendix A. Intuitively, for any transcript generated by Algorithm 1, if the "likelihood" is never exponentially high, then the transcript is $O(K/\sqrt{T})$-calibrated ($c$ will be a bounded constant). When $T$ is large, the calibration error will also go to 0 at a rate of $O(1/\sqrt{T})$. On the other hand, if the likelihood is exponentially large, we are "almost" perfectly predicting the label, so there is little uncertainty. Uncertainty quantification metrics such as calibration become less relevant.

### 3.3 IMPROVING STABILITY WITH PID CONTROL

Finally we address the stability problem. As we argued, the key problem is the constant need to increase or decrease the adjustment $E_{k+1}^t$ based on whether the empirical frequency $F_k^t$ is different from $\alpha_k t$. Our key insight is to think of this problem as a control problem: we would like $F_k^t$ to match $\alpha_k t$ and the error is measured by $E_k^t$ (which we want to bring close to 0), so we can use control algorithms to stabilize the system. We show (heuristically) that a very popular control algorithm called proportional-integral-derivative (PID) is well suited for the task.

Our new algorithm replaces $E_k^t$ in line 4 of Algorithm 1 with the weighted sum of three terms: **proportional** $E_{pk}^t = E_k^t$, **integral** $E_{ik}^t = \sum_{\tau=0}^t E_k^\tau$ and **derivative** $E_{dk}^t = E_k^t - E_{k-1}^t$. The proportional term $E_{pk}^t$ is the same as Eq.(3), so if we only have the proportional term we would reduce to Algorithm 1. The integral term $E_{ik}^t$ is key to addressing the stability problem because it does not decrease even when the $E_k^t$ goes to 0, so it applies a permanent adjustment when distribution drifts. The intuition of the integral term is demonstrated in Figure 2. Using the integral term is prone

to a well known problem called overshoot. Intuitively, the integral term will increase until $E_k^t$ goes to 0, at which point the (permanent) adjustment might already be too large. The standard method to reduce overshoot is to use a derivative term $E_{dk}^t$ (Minorsky, 1922), and we illustrate its effect in Figure 2. The overall adjustment is equal to

$$E_{\text{PID}-k}^t = k_p E_{pk}^t + \text{CLIP}_{[-B,B]} \left( k_i E_{ik}^t + k_d E_{dk}^t \right) \tag{6}$$

where $k_p, k_i, k_d \geq 0$ are the hyper-parameters (again, we will show in Section 3.4 that choosing these hyper-parameters is not difficult). We also clip the integral and derivative terms to be within $[-B, B]$ to ensure Corollary 1 even in the worst case scenario. For any real data in our experiments, the clipping never took effect.

We show that applying PID control does not break the calibration guarantee in Theorem 2.

**Corollary 1** (Simplfied). *Under the same condition as Theorem 2, for any $k_p, k_i, k_d \geq 0$, Algorithm 1 with $E_k^t$ replaced by $E_{\text{PID}-k}^t$ in Eq.(6) is b-calibrated where $b = O(K/\sqrt{T} + c/T)$ on any transcript such that $\forall t, L(Z^t, Y^t) \leq e^{\Theta(c)}$.*

A non-asymptotic version is also available in the appendix. With PID control Corollary 1 is essentially identical to Theorem 2 (in fact exactly the same for the simplified asymptotic version). Therefore, PID improves stability without degrading the calibration guarantees.

### 3.4 HYPER-PARAMETER SELECTION

Our algorithm consists of hyper-parameters which on first sight, seem difficult to choose for a practitioner. Specifically, the hyper-parameters are $\beta, \delta$ in Eq.(3), which specify how large is the adjustment $E_k^t$; the "elastic coefficient" $\eta$ in the definition of $A_k^t$ and $B_k^t$, and the PID coefficients $k_p, k_i, k_d$.

Fortunately, we show that our algorithm is very robust to hyper-parameter choices. In fact, we tune these parameters on a few simple synthetic examples (similar to Figure 1), and fix them throughout all the experiments on real data. We show (in Figure 7 in the appendix) that even if we were to finetune these hyper-parameters specifically for each dataset, we would only see marginal improvements compared to using fixed hyper-parameters. Therefore, any practitioner can simply use the fixed hyper-parameters that we recommend, and generally do not have to worry about tuning them in practice.

## 4 RELATED WORK

In the regression setup, there are several notions of calibration. The most common one (which is studied in our paper) is probabilistic calibration (Gneiting et al., 2007; Kuleshov et al., 2018; Vovk et al., 2020). Other notions include marginal calibration (Gneiting et al., 2007), individual calibration (Zhao et al., 2020), distribution calibration (Song et al., 2019), threshold calibration, etc. Though not considered in our paper, calibration has also been extensively studied in the classification setup, calibration has been studied under (Brier, 1950; Murphy, 1973; Dawid, 1984; Cesa-Bianchi & Lugosi, 2006; Platt et al., 1999; Zadrozny & Elkan, 2001; 2002; Niculescu-Mizil & Caruana, 2005; Guo et al., 2017; Lakshminarayanan et al., 2017). There are also other notions of calibration such as multi-calibration (Hébert-Johnson et al., 2018), decision calibration (Zhao et al., 2021), classwise calibration (Kull et al., 2019), etc.

Many applications only require a confidence interval. For confidence intervals a notion very similar to calibration is coverage, i.e. the label should belong to the interval with high probability. Conformal prediction methods are the gold standard for predicting intervals with high coverage. (Vovk et al., 2005; Shafer & Vovk, 2008; Papadopoulos, 2008; Romano et al., 2019).

The above research primarily focus on the i.i.d. setup (or exchangeable setup), while many real world data are not i.i.d. In the online setup with distribution shift, regret minimization algorithms can guarantee asymptotic calibration (Cesa-Bianchi & Lugosi, 2006; Kuleshov & Ermon, 2017) for classification. For conformal prediction, when the likelihood ratio of the distribution shift is known, (Tibshirani et al., 2019) can still guarantee coverage. A concurrent work (Gibbs & Candès, 2021) can guarantee asymptotic coverage for conformal confidence intervals. Our work differs in that we focus on calibration, and solve additional feasibility and stability challenges.

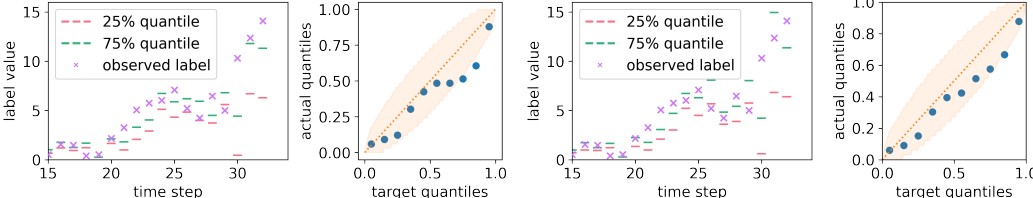

Figure 3: Example predictions on the stock prediction dataset (for Amazon). Left is conformal calibration only, and right is conformal calibration + our method. We plot both the sequence of predictions, and the reliability diagram (i.e. the relationship between target frequency $\alpha_k$ and empirical frequency $F_k^T/T$). Amazon growth has been faster than expected by analysts at times (such as steps 21-26, or 30-33), conformal calibration failed to capture this, while our method rapidly adapted to this distribution shift within a few time steps.

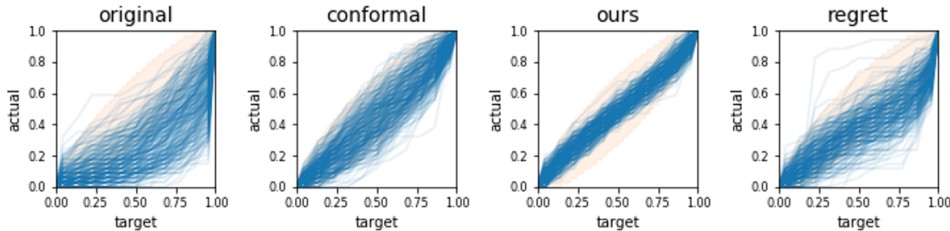

Figure 4: Reliability diagram for different methods on stock prediction. Each blue line is the reliability diagram for a single stock. The yellow shaded area is the 99% confidence interval for the reliability diagram of an oracle predictor (i.e. it is $\alpha_k \pm b_\delta^*(T)$ for $\delta = 0.01$). From left to right we use original predictions, conformal calibration, our method, and an regret minimization baseline. Our method achieves achieves the best calibration. Even though the regret minimization baseline can also guarantee calibration asymptotically, for short data sequences the calibration is incorrect.

## 5 EXPERIMENTS

This experiment aims to show on real time series data and standard regression benchmarks that our algorithm achieves strong calibration, sharpness and proper scores.

**Datasets** We use two time series prediction tasks (stock and COVID) and a regression benchmark with 17 datasets. For times series we start from expert predictions (such as from stock analysts or COVID forecasting teams), use conformal calibration to transform the expert predictions, and finally apply our algorithm to adjust the output of conformal calibration. For the regression benchmark we split each dataset into train and test. We use the training set to learn a prediction function offline, and assume that the test data are available sequentially online. For each dataset we add the following types of drifts: **linear shift**: add a constant bias to the label that increases over time; **cycle**: add a bias to the labels that moves up or down repeatedly; **scale shift**: multiple the label by a constant that increases with time; **jump**: add a constant bias at some $\forall t > t_0$ for some time step $t_0$. More details about the datasets are provided in appendix.

**Performance Metrics** : **Calibration loss** We use a notion similar to the expected calibration error (Guo et al., 2017) $\mathcal{L}_{\text{ece}} = \frac{1}{K}\sum_k |F_k^T - \alpha_k T|$. **Pinball loss** We use the average pinball loss (aka. hinge loss) $\mathcal{L}_{\text{pinball}}(Y, Z) = \frac{1}{K}\sum_k \alpha_k \text{ReLU}(Z - Y_k) + (1 - \alpha_k)\text{ReLU}(Y_k - Z)$. The pinball loss is a proper score, i.e. the oracle quantile prediction minimizes the expected pinball loss. **Sharpness** We use the average interval size to measure the sharpness $\mathcal{L}_{\text{sharpness}}(Z) = \frac{1}{K}\sum_k |Z_k - Z_{K-k}|$.

**Baselines**: We compare three baselines. **original** is the original expert predictions. We use kernel density estimation (Van Kerm, 2003) to transform the expert predictions into a probability; **conformal** is conformal calibration only (without Algorithm 1); **regret** is an adaptation of the regret minimization algorithm in (Kuleshov & Ermon, 2017) (details in the appendix).

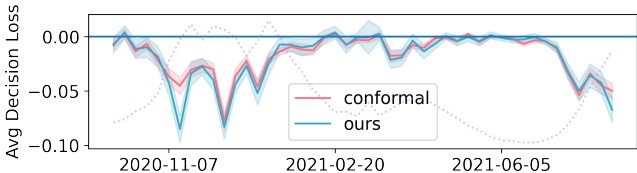

Figure 5: The decision loss reduction if we use conformal / ours instead of the original expert predictions for a COVID response decision task. The shaded area is 1 standard deviation computed by bootstrap sampling geographical locations, and the dashed line is the total case number. Both conformal / ours reduce decision loss (so all curves are below 0), but our method reduces loss more compared to conformal when distribution changes drastically, i.e. when case numbers spike.

| | pinball | | | | calibration (ece) | | | | sharpness | | | |
|---|---|---|---|---|---|---|---|---|---|---|---|---|
| | orig. | conf. | ours | regret | orig. | conf. | ours | regret | orig. | conf. | ours | regret |
| stock | 0.052 | **0.045** | 0.046 | 0.145 | 0.243 | 0.089 | **0.051** | 0.151 | 0.140 | 0.196 | 0.198 | 0.422 |
| COVID | 0.231 | **0.181** | **0.181** | 0.264 | 0.152 | 0.054 | **0.042** | 0.079 | 0.893 | 1.122 | 1.085 | 0.952 |
| Bench-shift | 2.540 | 0.921 | **0.299** | 0.505 | 0.494 | 0.436 | **0.149** | 0.232 | 0.676 | 2.842 | 1.737 | 2.768 |
| Bench-scale | 2.848 | 2.111 | **2.062** | 2.191 | 0.249 | 0.110 | **0.049** | 0.058 | 0.676 | 7.722 | 9.139 | 10.90 |
| Bench-jump | 1.396 | 0.777 | **0.533** | 0.762 | 0.286 | 0.201 | **0.073** | 0.172 | 0.676 | 2.164 | 1.972 | 2.637 |
| Bench-cycle | 0.924 | 0.697 | **0.496** | 0.714 | 0.220 | 0.165 | **0.047** | 0.146 | 0.676 | 2.595 | 2.775 | 3.045 |

Table 1: Performance comparison ours and several other alternatives on time series datasets and regression benchmarks. Our method (**ours**) achieves far better ECE than other methods. For pinball loss our method is on-par or better than baselines. The original prediction (**orig.**) is sharp, but the prediction is incorrect (the calibration and pinball loss are poor), hence sharpness is not meaningful. If we exclude (**orig.**), then our method achieves better or on-par sharpness than any other baseline. Regret minimization (**regret**) has poor performance on stock and COVID (with 50 samples per data sequence) and better performance on UCI (around 100 samples per data sequence) but still much worse than ours.

**Qualitative Results**   In Figure 3 and Figure 8,9 (in the Appendix) we plot some example prediction sequences and reliability diagrams. Our algorithm is able adjusts the predictions up or down to ensure calibration, and does not hurt sharpness (compared to conformal calibration only) when there is no distribution drift. In Figure 4 we plot the reliability diagram for the stock dataset. Our method is always calibrated (i.e. within the tolerance shaded in yellow), conformal calibration is uncalibrated (without the tolerance) for about 10% of the samples, while the regret minimization baseline is generally uncalibrated (because all time series are very short with around 50 samples).

**Quantitative Results**   In Table 1 we show the performance of our algorithm compared to several baselines. Our method achieves the best calibration error (ece) by far, and comparable or better pinball loss compared to baselines. Even though the original prediction achieves the best sharpness overall, the pinball loss and calibration error of the original prediction are so large, that sharpness is meaningless. If we exclude the original prediction, then our method also achieves comparable or better sharpness than other methods that have reasonable calibration.

**Downstream Task**   For COVID we simulate a response strategy (i.e. whether to close businesses) based on forecasted cases. We use the median of the prediction to select the best action. For details see Appendix C.1. The results are shown in Figure 5. Both conformal calibration and our method improves decision loss compared to using the original prediction; our method has an edge when the distribution changes rapidly, e.g. when cases surge.

## 6   CONCLUSION

This paper introduces a novel type of online algorithms to ensure calibration, where we start with an algorithm that guarantees calibration in the i.i.d. setup, and make modifications whenever it fails to achieve calibration. Future work can explore its application to other calibration definitions or more generally, other online learning problems.

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

## A    PROOFS

**Theorem 1.** *Let $\tilde{Z}_k^t$ be generated by any forecasting algorithm and bounded in $[-B, B]$, for any $\delta > 0$, the prediction algorithm defined by Eq.(3) is b-calibrated where $b(T) = b_\delta^*(T) + \frac{\log(2B+1)+\beta}{\beta T}$.*

*Proof of Theorem 1.* We prove that at any time $t$ we cannot have $|F_k^t - \alpha_k t| > b_\delta^*(t) + \log(1 + 2B)/\beta + 1$. This is equivalent to two statements that we prove separately by induction

$$F_k^t \geq \alpha_k t - b_\delta^*(t) - \log(1 + 2B)/\beta - 1 \tag{7}$$

$$F_k^t \leq \alpha_k t + b_\delta^*(t) + \log(1 + 2B)/\beta + 1 \tag{8}$$

To prove Eq.(7) suppose at time $t$ we have $F_k^t \geq \alpha_k t - b_\delta^*(t) - \log(1 + 2B)/\beta - 1$, then there are two situations:

Case 1. $F_k^t \geq \alpha_k t - b_\delta^*(t) - \log(1 + 2B)/\beta$, then we naturally have

$$F_k^{t+1} \geq F_k^t \geq \alpha_k t - b_\delta^*(t) - \log(1 + 2B)/\beta > \alpha_k(t+1) - b_\delta^*(t+1) - \log(1 + 2B)/\beta - 1 \tag{9}$$

Case 2. $F_k^t < \alpha_k t - b_\delta^*(t) - \log(1 + 2B)/\beta$, then by Eq.(3) we have

$$E_k^{t+1} > 1 + 2B - 1 = 2B$$

By the boundedness assumption,

$$Z^{t+1} \leq B < -B + E_k^{t+1} < \tilde{Z}_k^{t+1} + E_k^{t+1} = Y_k^{t+1}$$

so we must have $F_k^{t+1} = F_k^t + 1$, so

$$F_k^{t+1} = F_k^t + 1 \geq \alpha_k t - b_\delta^*(t) - \log(1 + 2B)/\beta > \alpha_k(t+1) - b_\delta^*(t) - \log(1 + 2B)/\beta - 1 \tag{10}$$

Similarly to prove Eq.(8) suppose at time $t$ we have $F_k^t \leq \alpha_k t + b_\delta^*(t) + \log(1 + 2B)/\beta + 1$ then at time $t + 1$ we must have

$$F_k^{t+1} \leq \alpha_k(t+1) + b_\delta^*(t+1) + \log(1 + 2B)/\beta + 1 \tag{11}$$

$\square$

**Theorem 2.** *Suppose the initial predictions are feasible (i.e. $\tilde{Z}_k^t < \tilde{Z}_{k+1}^t, \forall t, k$). For some $c > 2/\beta$, Algorithm 1 is b-calibrated where $b(T) = (2Kb_\delta^*(T) + Kc/T, 0)$ on any transcript that satisfies $\forall t \leq T, L(Y^t, Z^t) \leq \frac{1}{2\eta B(K+1)}(e^{\beta(c-1)/2} - 4B)$.*

*Proof of Theorem 2.* We first sketch our proof strategy. The proof will follow 3 steps:

1. We prove that if an interval contained the label too often, there will be a large force difference $E_{k+1}^t - E_k^t$.

2. We prove that if there is a large force difference, then the interval will be small.

3. If at time $t$ an interval contains the label $N$ times too often, at some previous time step $t' \leq t$, it must have contained the label $N - 1$ times too often and contained the label once again. However, at $t'$ the interval must have been small, hence the likelihood of the label must be large. Therefore, unless the likelihood is large, no interval can contain the label too often.

First we show a simple Lemma based on boundedness assumptions. This Lemma is a simple consequence of the boundedness $Y_0 = -B, Y_{K+1} = B$. We do not use this Lemma directly but it will be used in the proof of the other Lemmas.

**Lemma 1.** $\forall t, \forall k = 0, \cdots, K + 1, A_k^t < B, B_k^t > -B, |D_k^t| < B$

The following Lemma shows that intuitively, if the ground truth label belongs to an interval too often, then there $E_{k+1}^t$ must be significantly less than $E_k^t$

**Lemma 2.** *For any $c > 2/\beta$, if $F_{k+1}^t - F_k^t = \Delta\alpha_k t + 2b_\delta^*(t) + c$ then $E_{k+1}^{t+1} - E_k^{t+1} < -e^{\beta c/2}$.*

Next we show that if the forces $E_{k+1}^t$ is significantly smaller than $E_k^t$, then intuitively, this difference in force will "compress" the interval, making it small.

**Lemma 3.** *For any $L > 0$, if $E_{k+1}^t - E_k^t < -L$ we have $\Delta\tilde{Z}_k^t / \Delta Y_k^t > \frac{L - 4B}{2\eta}$.*

Based on these lemmas we can complete the proof. Suppose for some $k \in 1, \cdots, K$ and $c > 2$ we have $F_{k+1}^t - F_k^t = \Delta\alpha_k t + 2b_\delta^*(t) + c$, then consider last time that the label belongs to the interval, i.e. we want to find the largest $t' \le t$ such that $Z^{t'} \in [Y_k^{t'}, Y_{k+1}^{t'})$. For $t'$ we have

$$F_{k+1}^{t'-1} - F_k^{t'-1} = F_{k+1}^t - F_k^t - 1 \tag{12}$$

$$> \Delta\alpha_k t + 2b_\delta^*(t) + c - 1 \tag{13}$$

$$= \Delta\alpha_k t' + 2b_\delta^*(t') + (c + \Delta\alpha_K(t - t') + 2b_\delta^*(t) - 2b_\delta^*(t') - 1) \tag{14}$$

$$\ge \Delta\alpha_k t' + 2b_\delta^*(t') + c - 1 \tag{15}$$

Where the last inequality is because $b_\delta^*$ is monotonically non-decreasing in $t$. Therefore by Lemma 2 we have

$$E_{k+1}^{t'} - E_k^{t'} < -e^{\beta(c-1)/2} \tag{16}$$

So by Lemma 3 we can conclude

$$\Delta\tilde{Z}_k^{t'} / \Delta Y_k^{t'} > \frac{e^{\beta(c-1)/2} - 4B}{2\eta} \tag{17}$$

which implies whenever $e^{\beta(c-1)/2} > 4B$, or $c > 2\log(4B)/\beta + 1$

$$\Delta Y_k^{t'} < 2\eta\Delta\tilde{Z}_k^{t'}(e^{-\beta(c-1)/2} - 4B) < 2\eta B(e^{-\beta(c-1)/2} - 4B) = 2\eta B(e^{-\beta(c-1)/2} - 4B) \tag{18}$$

Hence. by the definition of the "likelihood"

$$L(Y^{t'}, Z^{t'}) > \frac{\alpha_{k+1} - \alpha_k}{2\eta B}(e^{\beta(c-1)/2} - 4B) = \frac{1}{2\eta B(K+1)}(e^{\beta(c-1)/2} - 4B) \tag{19}$$

In other words, unless $\exists t'$ such that $L(Y^{t'}, Z^{t'}) > \frac{1}{2\eta B(K+1)}(e^{\beta(c-1)/2} - 4B)$, we cannot have $F_{k+1}^t - F_k^t = \Delta\alpha_k t + 2b_\delta^*(t) + c$.

Finally we connect the bound on $F_{k+1}^t - F_k^t$ with the quantile calibration error.

**Lemma 4.** *For any $b > 0$, if $\forall k \in K$, $F_{k+1}^t - F_k^t \le \Delta\alpha_k t + b$, then*

$$\max_k |F_k^t - \alpha_k t| \le Kb \tag{20}$$

So combined we have if $\forall t, L(Y^t, Z^t) \le \frac{1}{2B(K+1)}(e^{(c-1)/2} - 4B)$, then

$$\max_k |F_k^t - \alpha_k t| \le 2Kb_\delta^*(t) + Kc \tag{21}$$

Finally we prove the lemmas used in this theorem.

*Proof of Lemma 2.* Suppose $F_{k+1}^t - F_k^t = \Delta\alpha_k t + 2b_\delta^*(t) + c$ We first use simple algebraic transformation to get

$$(F_{k+1}^t - \alpha_{k+1}t) - (F_k^t - \alpha_k t) = c + 2b_\delta^*(t) \tag{22}$$

For notation simplicity denote the first term LHS as $u = F_{k+1}^t - \alpha_{k+1}t$ and second term as $v = F_k^t - \alpha_k t$, we can simplify the equation as so $u - v = 2b_\delta^*(t) + c$.

First we consider the scenario where the interval is not the first interval $\alpha_0, \alpha_1$ or the last interval $\alpha_K, \alpha_{K+1}$, in other words, we consider the situation where $k \in [1, K-1]$. There are three different

possible situations: $u \geq 0, v \geq 0$ and $u \geq 0, v < 0$ and $u < 0, v < 0$. Note that we can never have $u < 0, v \geq 0$. We separately consider the three situations.

1. If $u \geq 0, v \geq 0$ then

$$E_{k+1}^{t+1} - E_k^{t+1} = -e^{\beta(u-b_\delta^*(t))} + 1 - (-e^{\beta(v-b_\delta^*(t))} + 1) \tag{23}$$

$$= -e^{\beta(u-b_\delta^*(t))} + e^{\beta(v-b_\delta^*(t))} \leq 1 - e^{\beta(c+b_\delta^*(t))} \tag{24}$$

where the final inequality achieves equality when $u = c + 2b_\delta^*(t), v = 0$

2. If $u \geq 0, v < 0$ then

$$E_{k+1}^{t+1} - E_k^{t+1} = -e^{\beta(u-b_\delta^*(t))} + 1 - (e^{-\beta(v+b_\delta^*(t))} - 1) \tag{25}$$

$$= -e^{\beta(u-b_\delta^*(t))} - e^{-\beta(v+b_\delta^*(t))} + 2 < -2e^{\beta c/2} + 2 \tag{26}$$

where the final inequality achieves equality when $u = (c + 2b_\delta^*(t))/2, v = -(c + 2b_\delta^*(t))/2$.

3. If $u < 0, v < 0$ then

$$E_{k+1}^{t+1} - E_k^{t+1} = (e^{-\beta(u+b_\delta^*(t))} - 1) - (e^{-\beta(v+b_\delta^*(t))} - 1) \tag{27}$$

$$= e^{-\beta(u+b_\delta^*(t))} - e^{-(v+b_\delta^*(t))} \leq 1 - e^{\beta(c+b_\delta^*(t))} \tag{28}$$

where the final inequality achieves equality when $u = 0, v = -c - 2b_\delta^*(t)$.

Next we consider the situation where $k = 0$. In this case $v = F_0 - \alpha_0 t = 0$ by assumption. So

$$E_1^{t+1} - E_0^{t+1} = E_1^{t+1} = -e^{\beta(c+2b_\delta^*(t)-b_\delta^*(t))} + 1 = -e^{\beta(c+b_\delta^*(t))} + 1 \tag{29}$$

Next we consider the situation where $k = K$. Similar to the previous argument, $u = 0$ by assumption, so

$$E_{K+1}^{t+1} - E_K^{t+1} = -E_K^{t+1} = -(e^{\beta(c+2b_\delta^*(t)-b_\delta^*(t))} - 1) = 1 - e^{\beta(c+b_\delta^*(t))} \tag{30}$$

Combined we have $\forall k$, we take the maximum of the lower bounds and have

$$E_{k+1}^{t+1} - E_k^{t+1} < \max(-2e^{\beta c/2} + 2, 1 - e^{\beta(c+b_\delta^*(t))}) \tag{31}$$

Since $c > 2/\beta$ we can further simplify this expression by observing $-2e^{\beta c/2} + 2 < -e^{\beta c/2}$ and $1 - e^{\beta(c+b_\delta^*(t))} < 1 - e^{\beta c} < e^{\beta c/2}$, so we have

$$E_{k+1}^{t+1} - E_k^{t+1} < \max(-2e^{\beta c/2} + 2, 1 - e^{\beta(c+b_\delta^*(t))}) < -e^{\beta c/2} \tag{32}$$

$\square$

*Proof of Lemma 3.* Observe the equilibrium equation $D_k + B_k + A_k + E_k = 0$ and $D_{k+1} + B_{k+1} + A_{k+1} + E_{k+1} = 0$; by Lemma 1

$$A_k < B, B_k > -B, |D_k| < B$$

so consequently we have

$$A_k - B_{k+1} = D_{k+1} + E_{k+1} + B_{k+1} - D_k - E_k - A_k < -L + 4B$$

By the definition of $A_k$ and $B_{k+1}$ we have

$$2\eta \left( \frac{\Delta Y_k}{\Delta \tilde{Z}_k} - \frac{\Delta \tilde{Z}_k}{\Delta Y_k} \right) < -L + 4B \tag{33}$$

which implies that

$$\frac{\Delta \tilde{Z}_k}{\Delta Y_k} > \frac{L - 4B}{2\eta} + \frac{\Delta Y_k}{\Delta \tilde{Z}_k} > \frac{L - 4B}{2\eta} \tag{34}$$

$\square$

*Proof of Lemma 4.* We prove the contra-positive statement of the Lemma. If $\max_k |F_k^t - \alpha_k t| \leq Kb$ is violated, there must be one of two situations:

1. there exists some $k$ such that $F_k^t - \alpha_k t \geq Kb$ then

$$Kb \leq F_k^t - \alpha_k t - F_0^t + \alpha_0 t \qquad\qquad F_0^t \equiv 0, \alpha_0 t \equiv 0 \qquad (35)$$

$$= \sum_{0 \leq j \leq k} (F_j^t - F_{j-1}^t) - (\alpha_j - \alpha_{j-1})t \qquad\qquad \text{Telescope} \qquad (36)$$

$$\leq (k+1) \max_{j \leq k} (F_j^t - F_{j-1}^t) - (\alpha_j - \alpha_{j-1})t \qquad\qquad (37)$$

2. there exists some $k$ such that $F_k^t - \alpha_k t \leq -b$ then

$$Kb \leq -F_k^t + \alpha_k t + F_{K+1}^t - \alpha_{K+1} t \qquad\qquad F_{K+1}^t \equiv t, \alpha_{K+1} t \equiv t \qquad (38)$$

$$= \sum_{k \leq j \leq K+1} (F_j^t - F_{j-1}^t) - (\alpha_j - \alpha_{j-1})t \qquad\qquad \text{Telescope} \qquad (39)$$

$$\leq (K+1-k) \max_{j \geq k} (F_j^t - F_{j-1}^t) - (\alpha_j - \alpha_{j-1})t \qquad\qquad (40)$$

In both cases, there must be some $j$ such that

$$(F_j^t - F_{j-1}^t) - \Delta \alpha_{j-1} t \geq b \qquad\qquad (41)$$

$\square$

$\square$

**Corollary 1.** Suppose the initial predictions are feasible (i.e. $\tilde{Z}_k^t < \tilde{Z}_{k+1}^t, \forall t, k$). For some $c > 2/\beta$, the prediction algorithm in Section 3.3 is $b$-calibrated where $b(T) = 2Kb_\delta^*(T) + Kc/T$ on any transcript that $\forall t, L(Y^t, Z^t) \leq \frac{1}{2\eta B(K+1)} (e^{\beta(c-1)/2} - 6B)$.

*Proof of Corollary 1.* To prove this corollary we only have to modify Lemma 2, and the rest of the proof follows as Theorem 2.

**Lemma 5.** *For any* $c > 2\beta$, *if* $F_{k+1}^t - F_k^t = \Delta\alpha_k t + 2b_\delta^*(t) + c$ *then* $E_{\text{PID}-k+1}^{t+1} - E_{\text{PID}-k}^{t+1} < -e^{\beta c/2} + 2B$.

Lemma 5 holds because $E_{\text{PID}-k}^{t+1}$ differs from $E_k^{t+1}$ by at most $B$, so

$$E_{\text{PID}-k+1}^{t+1} - E_{\text{PID}-k}^{t+1} < E_{k+1}^{t+1} - E_k^{t+1} + 2B < -e^{\beta c/2} + 2B \qquad\qquad (42)$$

$\square$

# B  ADDITIONAL EXPERIMENT RESULTS

## B.1  HYPER-PARAMETER TUNING

We tune hyper-parameters that minimize a weighted combination of the expected calibration error (ece) and the pinball loss.

$$\mathcal{L}_{\text{mix}}(w_{\text{mix}}) = w_{\text{mix}}\mathcal{L}_{\text{ece}} + (1 - w_{\text{mix}})\mathcal{L}_{\text{pinball}} \qquad\qquad (43)$$

where $w_{\text{mix}}$ is a weight coefficient we choose to be 0.4. We use Optuna (Akiba et al., 2019) to seek promising hyper-parameter candidates. We try 100 hyper-parameter candidates and pick the best hyper-parameter that minimizes the loss function $\mathcal{L}_{\text{mix}}(w_{\text{mix}})$.

As we mentioned in Section 3.4, we use simple synthetic tasks for tuning. We first define the label of a task as follows:

$$Y^t = a + b\epsilon^t, \ \epsilon^t \sim \mathcal{N}(0, 1) \ (t = 1, \ldots, T) \qquad\qquad (44)$$

| | weight ($w_{\mathrm{mix}}$) | | | | | |
| --- | --- | --- | --- | --- | --- | --- |
| | 0.0 | 0.2 | 0.4 | 0.6 | 0.8 | 1.0 |
| pinball | 0.155 | 0.155 | 0.159 | 0.155 | 0.166 | 0.178 |
| calibration(ece) | 0.021 | 0.015 | 0.009 | 0.022 | 0.007 | 0.008 |
| sharpness | 0.811 | 0.826 | 0.859 | 0.812 | 0.921 | 0.972 |

Table 2: Comparison of performance metrics between weight $w_{\mathrm{mix}}$.

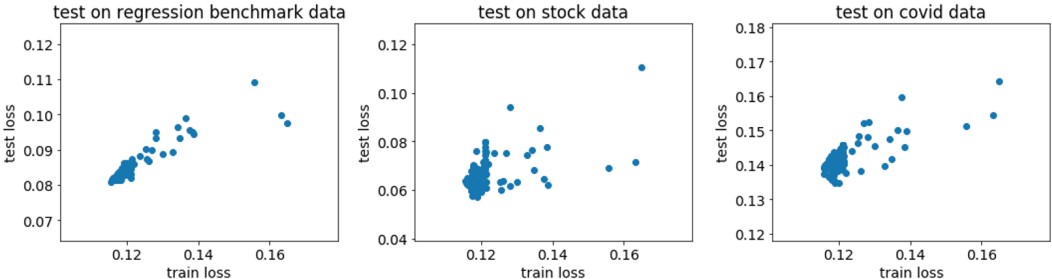

Figure 6: Comparison of loss values between synthetic data and other datasets, where each point corresponds to a hyper-parameter candidate which is obtained during hyper-parameter search. The $x$-axis and $y$-axis show loss values on synthetic dataset and other datasets, respectively. The performance on simple synthetic data is highly correlated with the performance on real data.

where coefficients $a$ and $b$ are sampled by $a \sim \mathcal{N}(0,1)$ and $b \sim U(0,1)$. We also set predictions $\tilde{Z}^t = F^{-1}(\alpha_k)$ where $F$ is the cumulative distribution function of standard Normal distribution.

After tuning, we evaluate the best hyper-parameters for each $w_{\mathrm{mix}}$ on other synthetic tasks. Table 2 compare performance metrics between different $w_{\mathrm{mix}}$. Large $w_{\mathrm{mix}}$ leads to small calibration loss values and large pinball loss/sharpness values. We adopt $w_{\mathrm{mix}} = 0.4$ since the hyper-parameter provides good calibration and pinball values. The obtained hyper-parameter values are $\delta = 0.47, \beta = 0.16, k_d = 0.08, k_p = 1.00, \eta = 0.96$. As for $k_i$, we assume $k_i$ depends on quantile level $k$ as follows:

$$k_{i,k} = k_{i,\max} - (k_{i,\max} - k_{i,\min})|1 - 2(k-1)/(K-1)|, \ \forall 1 \le k \le K. \tag{45}$$

The obtained hyper-parameters are $k_{i,\max} = 0.09, k_{i,\min} = 0.04$.

Next we verify that our algorithm is robust to hyper-parameter choice. In Figure 6 we compare the performance of a hyper-parameter on synthetic sequences and real data. They are highly correlated, which indicates that hyper-parameters that do well on synthetic sequences also perform well on real data.

Finally, we demonstrate that the adopted hyper-parameter in the paper works well compared to "optimal" hyper-parameters for each dataset. Concretely, we tune hyper-parameter on regression benchmark datasets with distribution drifts, and compare the tuned hyper-parameter and the recommended ones (which are tuned on synthetic sequences). Figure 7 shows that tuning on each dataset separately (which is only possible for off-line data) only improves performance marginally compared to using fixed hyper-parameters. This suggests that the recommended hyper-parameter is a reasonable choice in a variety of cases.

## B.2 VISUALIZATIONS OF PREDICTIONS ON VARIOUS DATASETS

In addition to Figure 3 in Section 5, we visualize additional example predictions on COVID and regression benchmark datasets. Figure 8 shows example predictions on the COVID prediction dataset, and Figure 9 shows example predictions on regression benchmark datasets with distribution drifts. Our method adapts to distribution drifts.

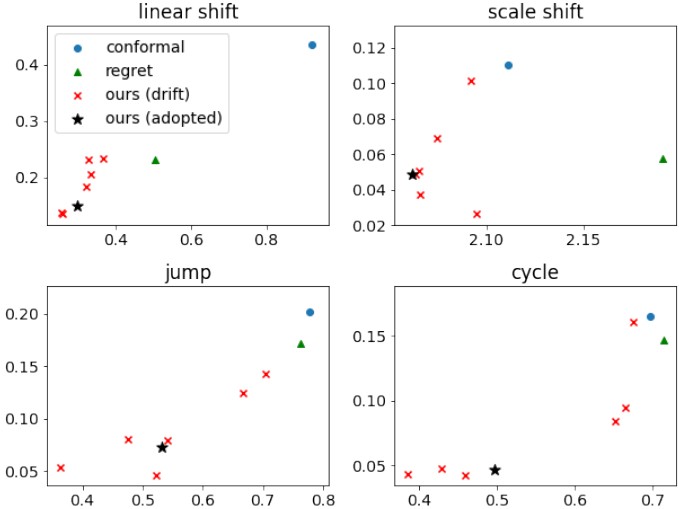

Figure 7: Evaluating the robustness to hyper-parameter choice. The $x$-axis and $y$-axis shows the pinball loss and ece, respectively. The star is the hyper-parameter we fix throughout the experiments. The red crosses are the hyper-parameters that are optimized on regression benchmark datasets for each distribution drift type respectively. Each red cross in a figure corresponds to different weight $w_{\mathrm{mix}}$. Even though optimizing hyper-parameters for each dataset separately can further improve performance, the improvement is minor.

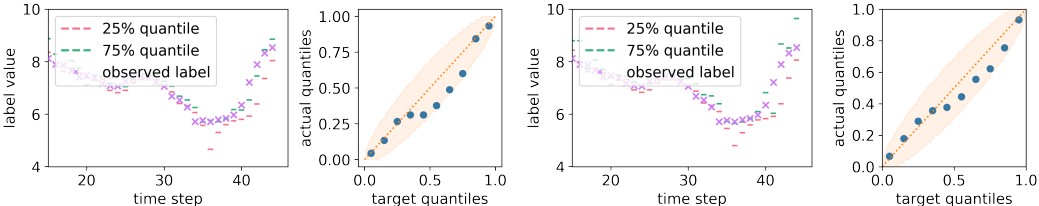

Figure 8: Example predictions on the COVID prediction dataset. Left is conformal calibration only, and right is conformal calibration + our method. We plot both the sequence of predictions, and the reliability diagram (i.e. the relationship between target frequency $\alpha_k$ and empirical frequency $F_k^T/T$).

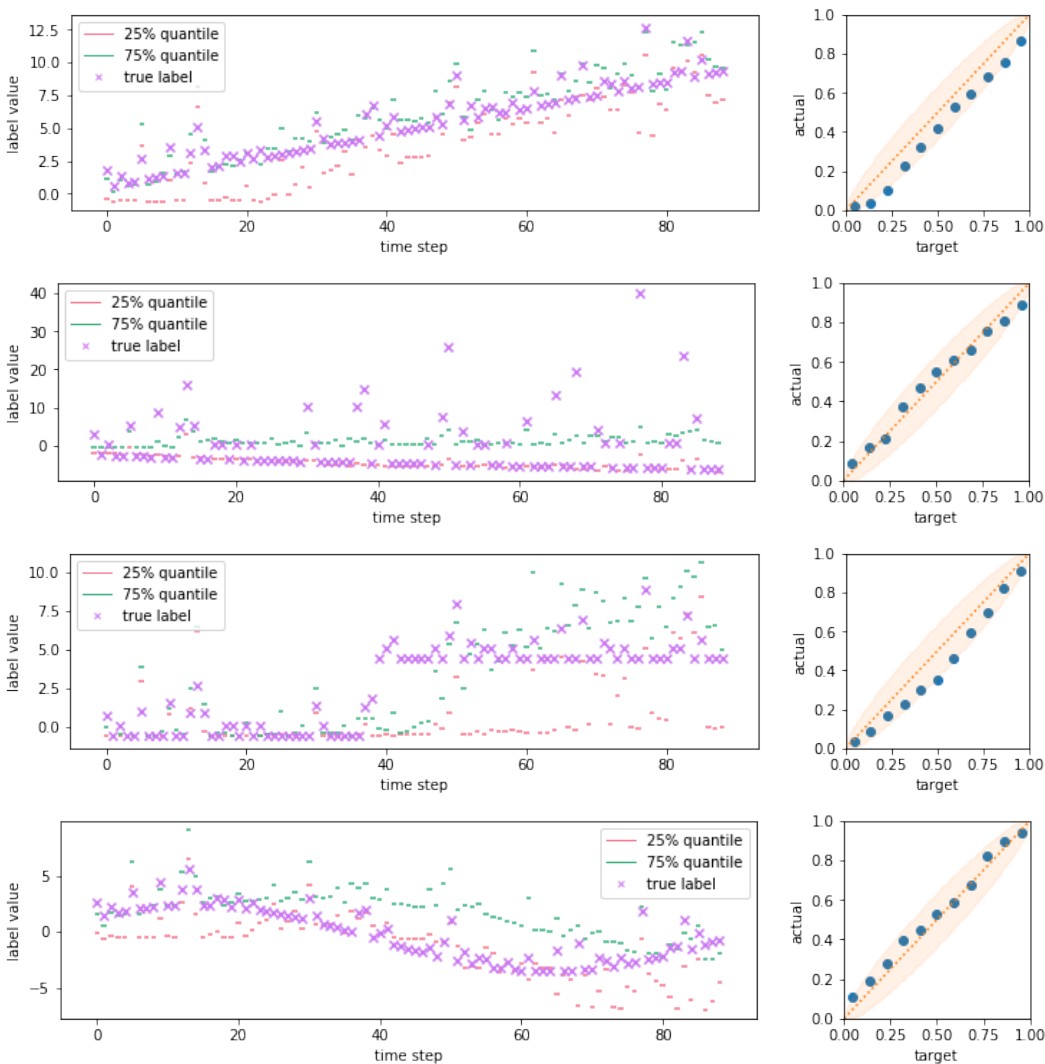

Figure 9: Example predictions on regression benchmark datasets with distribution drifts. From top to bottom, we apply 'linear shift', 'scale shift', 'jump', and 'cycle', respectively. We plot both the sequence of predictions, and the reliability diagram (i.e. the relationship between target frequency $\alpha_k$ and empirical frequency $F_k^T/T$).

## C  EXPERIMENT DETAILS

**Stock Dataset**  For stock we predict quarterly earnings per share (EPS) for S&P 500 stocks from 2010/01 to 2021/03. We use analyst earnings forecasts provided by I/B/E/S as the original prediction. To remove stocks with low quality data, we apply the following filtering rule for each stock.

1. exclude analyst forecasts whose missing rates are over 20%
2. if the number of analysts is less than five, remove the stock
3. if all remaining analysts do not predict earnings at least a quarter, remove the stock

By the above procedure, 279 stocks are selected from 500 stocks in S&P 500. The average number of analysts per stocks is 9.3 and each time series length is 45 data points.

**COVID dataset**  For COVID we predict COVID cases for the next week. We use expert forecasts from the COVID forecast hub (Ray et al., 2020). Because the data is heavy tailed, we transform all forecasts and labels by $Z \mapsto \log(1 + Z)$ and predict the transformed label. We select the teams and locations that do not have missing predictions during the period between Aug 2020 and Aug 2021 (6 teams) as the original prediction, and randomly sample 100 locations. The time series length is 56 data points (so approximately one year of data).

**Regression Benchmarks**  We use the following UCI datasets: blog, boston, concrete, crime, energy-efficiency, fbcomment-1, fbcomment-2, forest-fires, mpg, naval, power-plant, protein, superconductivity, wine, and yacht. In addition to UCI datasets we also use two common regression datasets in the uncertainty quantification literature, which are kin8nm and medical-expenditure. We use 100 samples for each test dataset, i.e., the length is 100.

**Distribution drifts**  For the regression benchmarks we consider four types of artificial distribution drifts. The definition of the distribution drift is given by

- linear shift: add linear drifts to the labels, i.e., $Y^t = Y^t + \beta t$
- scale shift: change the scales of the labels, i.e., $Y^t = Y^t(1 + \beta\sqrt{t})$
- jump: jump the mean at the halfway point, i.e., $Y^t = Y^t \ (t < T/2), \ Y^t = Y^t + \beta \ (t \geq T/2)$
- cycle: add fluctuations to the labels, i.e., $Y^t = Y^t + \beta \sin(2\pi t/T)$

In the experiments, we set coefficient $\beta = 0.1, 1.0, 3.0, 3.0$ for linear shift, scale shift, jump, and cycle, respectively.

### C.1  DOWNSTREAM TASK DETAILS

**COVID Decision Making**  We consider the following simple decision loss: for each county, there are four tiers of response 0,1,2,3. A higher tier response is needed whenever there are more forecasted cases. Let $a$ denote the response, and $z$ denote the (log) cases, we choose the loss function as the sum of COVID losses and shut-down losses, respectively defined as

|  |  | $a = 0$ | $a = 1$ | $a = 2$ | $a = 3$ |
|---|---|---|---|---|---|
| COVID loss | $z \leq 4$ | 0 | 0 | 0 | 0 |
|  | $4 < z \leq 6$ | 1 | 0 | 0 | 0 |
|  | $6 < z \leq 8$ | 3 | 2 | 0 | 0 |
|  | $z > 8$ | 6 | 5 | 3 | 0 |
| Shutdown loss |  | 0 | 0.3 | 0.7 | 1.0 |

### C.2  ONLINE LEARNING BASELINE

We adopt Algorithm 1 in (Kuleshov & Ermon, 2017) based on internal regret minimization, we use exactly the same algorithm except that (Kuleshov & Ermon, 2017) minimizes the L2 loss or Brier loss (which are proper scoring rules for the conditional expectation), while we minimize the

pinball loss (which is the proper scoring rule for quantiles). In addition, we separately calibrate the different quantiles $Y_1, \cdots, Y_K$ (i.e. for each $k$ we run a separate instance of modified Algorithm 1 in (Kuleshov & Ermon, 2017)).

The original procedure in (Kuleshov & Ermon, 2017) can guarantee calibration in the binary classification setting; we give intuition on why our modification can guarantee calibration in the regression setting. we first bin the possible regression outputs into $M$ equal sized bins $-B = y_0 < \cdots < y_M = B$. As a notation convenience, denote the center of each bin as $y_m^c = \frac{y_m + y_{m-1}}{2}$.

Consider the $\alpha_k$-th quantile, our goal is to minimize the interval regret under the pinball loss, define by $\forall c \in [-B, B]$

$$R_{\text{interval}}^c = \sum_t \mathbb{I}(Y_k^t = c)\mathcal{L}_{\text{pinball}}(c, Z^t) - \inf_y \sum_t \mathbb{I}(Y_k^t = c)\mathcal{L}_{\text{pinball}}(y, Z^t)$$

Intuitively, among the time that the true label equals $c$ there is not a better alternative prediction $y$ that achieves lower pinball loss. Because pinball loss is a proper scoring rule, $R_{\text{interval}}^c = 0$ implies that $c$ is the $\alpha_k$-quantile of $\{Z^t \mid Y_k^t = c\}$, i.e. for this subset of samples, the true label $Z^t$ is below the prediction $Y_k^t = c$ for $\alpha_k$ proportion of times. This is exactly the definition of calibration. To implement the algorithm realistically, we have to choose a suitable binning strategy, as well as a internal regret minimization algorithm. We use the same choice as the experiments in (Kuleshov & Ermon, 2017). Note that to choose the binning strategy we have to know the lower bound and upper bound of the data. We give the baseline an unfair advantage by choosing the lower bound as the lowest value in the data, and the upper bound as the highest value in the data (up to a 10% margin).

