# OpenReview forum: "Provably Calibrated Regression Under Distribution Drift"
_ICLR.cc/2022/Conference — ICLR 2022 Submitted_

### Official Review · Reviewer_Kb8i · 2021-10-28

**Correctness:** 3
**Technical Novelty And Significance:** 3
**Empirical Novelty And Significance:** 2
**Recommendation:** 5
**Confidence:** 3

**Main Review:**

**Strengths**
1. calibration for online regression is an important problem and well-motivated.
2. The proposed algorithm is well-supported by theory.

**Concerns**
1. It is counter-intuitive that applying PID control does not break the calibration guarantee for any PID parameters; in control theory, given a plant (or model dynamics), only a few combinations of PID parameters can make the system stable. However, Corollary 1 states that the proposed algorithm is b-calibration for all PID parameters, which may need additional explanation. Could you clarify this?
2. I think the modeling choice (3) implicitly makes some assumptions on the underlying distribution drift, which is also assumptions on the theorems and corollary. In particular, (3) adjusts calibration values made by the iid algorithm (e.g., conformal calibration) with exponential models; this implies that the distribution shift may not be significant. Are you making assumptions on distribution drift? If so, could you clarify the underlying assumption on (3)?
3. Related to the above two points, it's not easy to believe that hyper-parameter tuning is easy; in particular, PID parameter tuning is generally known to be difficult without knowing model dynamics. I guess the reason that the hyper-parameter tuning is easy can be implicit assumptions on the distribution drift, but it's unclear as it is written now. Could you discuss convincing reasons that the tuning is easy (in particular the PID parameters)?
4. The connection between theory and empirical results is loose; is the proposed algorithm empirically b-calibrated? If so, could you add empirical results/evidences?
5. I believe that well-known and acceptable regression calibration approaches under iid (in particular [R1]) need to be compared; even though this paper address online regression, it's necessary to compare with the well-known and acceptable iid regression calibration algorithms to show the efficacy of the proposed approach under distribution drift (along with conformal calibration).
6. I think ablation study makes this paper stronger; could you add this (e.g., by adding "Basic + Feasibility" algorithm performance)?
7. other comments:
* "transforms a initial predictor" -> "transforms an initial predictor"
* What are "Bench-shift" and others? They are not defined.

[R1] https://arxiv.org/abs/1807.00263

**Summary Of The Paper:**

This paper consider calibrated online regression; to attack this problem, this paper proposes a new calibration algorithm (i.e., adjusting conformal calibration for distribution drift by ensuring feasibility and improving stability) and proves the proposed algorithm is "b-calibrated" (as defined in the paper). The proposed approach is evaluated two time series data (i.e., stock and COVID) and 17 regression datasets, and demonstrated that it outperforms three baselines in terms of two calibration metrics (i.e, pinball and ECE).

**Summary Of The Review:**

I think this paper considers an interesting and important problem; the proposed approach is novel, but it not well justified in theoretically and empirically as mentioned in the main review. I'm willing to adjust my understanding and score as well.

---

### Official Review · Reviewer_3r9U · 2021-10-30

**Correctness:** 3
**Technical Novelty And Significance:** 3
**Empirical Novelty And Significance:** 3
**Recommendation:** 5
**Confidence:** 4

**Main Review:**

This paper mainly focuses on the online setting with distribution drifts.
* For calibration, the conformal learning is used which is not new.
* For distribution drifts, the PID control is introduced with theoretical guarantee.
* The major concern lies in the experimental part. This paper considers four types of drifts but the the drift data are synthetic. Is there any real applications that needs distribution drifts?
* For baselines, is there any online variants of the existing methods? Although the current calibration/Bayesian methods are not directly available for distribution drifts, there should be some prompt apply and variants of these existing methods that need to be added as baselines.

**Summary Of The Paper:**

This paper proposes a online regression calibration method using conformal and control theories. Theoretical and experimental results are provided.

**Summary Of The Review:**

This paper considers an new but important branch of truthworthy regression/forecasting, online learning with distribution drifts. The draft is clearly written but some concerns lie in the novelty and experimental setting.

---

### Official Review · Reviewer_DfEt · 2021-11-02

**Correctness:** 2
**Technical Novelty And Significance:** 2
**Empirical Novelty And Significance:** 3
**Recommendation:** 5
**Confidence:** 3

**Main Review:**

While the authors try to explain intuition through several analogies from physics and dynamical systems, the paper is still sometimes hard to follow. I was wondering whether the authors could comment on the following questions:

--- The paper heavily refers to conformal calibrators. In the current form, it seems to be a bit hand-wavy, so a slightly more detailed description of the procedure (either in the main paper or in the Appendix) would be helpful to the reader.

--- The confidence interval (CI) statement at the beginning of section 3.1 seems to be a bit odd as usually CIs are constructed for unknown parameters, while here it is used for testing for the i.i.d. assumption (a side note: presumably $\geq$ sign should be used in place of equality). Continuing on the same part, shouldn't one look at confidence sequences instead of confidence intervals for the sample average of Bernoulli trials? That is, confidence intervals usually require pre-specified sample size to be valid (and thus are not fully applicable to sequential settings), while confidence sequences yield guarantees that are time-uniform, and thus allow for data-adaptive collection settings.

--- in the proof, it seems to be used that $b_\delta^*(t+1)>b_\delta^*(t)$, which is not discussed. Is it indeed the case? In general, slightly more details for this proof are needed (including the base case for the induction and more elaboration on omitted steps)

--- Solving the equilibrium equations (as in eq. 4). How is it done exactly? A bit of guidance on algorithmic aspects, computational complexities etc. could be helpful.

--- A side note about some typos: a) before stating equation (3), it is mentioned that eq. 3 states a prediction for $Y^t_k$. Shouldn't it be $Y^{t+1 }_k$? Similarly, just formally, shouldn't $Z^{t+1 }_k$ also be clipped in $[-B,B]$? b) A general suggestion could be to move the related work section to an earlier part of the paper for better readability.

**Summary Of The Paper:**

For real-world applications where ML models are used, quantifying predictive uncertainty is an essential task as, for example, in safety-critical applications misclassification might have disastrous consequences. On the theory side, the analysis is typically performed under the i.i.d. assumption which could be unsatisfactory as deployed models inevitably encounter changes in the data generating distribution and/or certain dependencies that invalidate the results established for the i.i.d. setting. Focusing on regression, the authors design an adaptive (to distribution shifts) procedure that satisfies a certain calibration guarantee. The authors study the empirical performance of the proposed procedure on a collection of time-series and regression datasets.

**Summary Of The Review:**

Lowering the score is mainly motivated by the issues that are discussed in the main review. Unless resolved, the general recommendation for the current work is that it is below the acceptance threshold.

---

### Official Review · Reviewer_Lt92 · 2021-11-03

**Correctness:** 2
**Technical Novelty And Significance:** 3
**Empirical Novelty And Significance:** 2
**Recommendation:** 5
**Confidence:** 3

**Main Review:**

This paper investigates the problem of calibrating regression outputs with a distribution drift setting. Given that existing regression calibration methods relies on the i.i.d samples to estimate the empirical quantiles, it is interesting to see how the authors would tackle the more complicated problem of a non-iid data source.

The author first introduced an easier requirement of online quantile calibration, where the calibration error can be accepted up to a given function $b(T)$ at time step $T$. While the author initially suggested the function $b$ can be arbitrary, it later shows that the function needs to be specified for the proposed calibration method.

The proposed solution is then based on an online adjustment scheme. While the intuition is straightforward (change the prediction value if it is outside of the current confidence interval), it is unclear whether the authors pick this particular exponential form to adjust the predicted quantile value. The adjustment scale is about $1$ and an exp term and not about the original scale of $Y$.

Later in theorem 1, the authors show that the previous methods can achieve the so-called b-calibration if the function $b$ is given in a particular form. Again, here the author didn't elaborate regarding the choice of $b$, and it is likely that it needs to be selected according to eq.3 (or vice versa) to ensure the theorem stands.

As the algorithms only work on each quantile, the authors later provide an adjustment method to ensure all the quantiles are monotonic and another PID-control based approach to improve the stability of quantiles.

Experiments are mainly conducted against the standard conformal calibration approach, and some improvements can be observed.

**Summary Of The Paper:**

This paper considers the problem of calibrating quantiles outputted by a regression model. The data is assumed to draw from a non-iid source (e.g. time series with distribution shift).
The paper proposes a method that adjusts the output quantiles to be better calibrated for a less-restricted definition of b-calibrated.
Some further methods are also proposed to ensure the quantiles are monotonic and stable.
Experiments are conducted to compare the proposed method with uncalibrated models and conformal predictions.

**Summary Of The Review:**

While this paper is working on a very attractive problem and provides a working solution, some technical decisions require further elaboration to make the paper technically stronger.
1. The choice of the $b$ function and adjustment in eq.3 needs to be further discussed. Otherwise, Theorem 1 seems to be solving a self-defined problem.
2. While the methods used to ensure monotonicity and stability seems to work well in the current experiments, it is unclear whether it will require such complicated steps to solve the issue. I wonder if a regularisation term plus isotonic regression on the $Z^t_{k}$ would also solve the problem while always ensuring monotonicity.
3. Improve model under distribution-shift is a well-established domain, I wonder if other reviewers with a stronger background in out-of-distribution prediction could enlighten if there are any good baseline to compare.

---

### Decision · Program_Chairs · 2022-01-20

**Decision:**

Reject

**Comment:**

The paper studies an important problem of quantifying uncertainty (as measure by calibration) of predictions made by an ML algorithm in the presence of distribution drift. However, all reviewers point out a slew of concerns that went un-rebutted by the authors. The reviewers concurred that the paper deserved to be rejected at the current stage, and I concur. I recommend that the authors take the critical and constructive feedback into account to improve the paper and perhaps resubmit to a different venue in 2022.